# Weaker cooling by aerosols due to dust-pollution interactions

Klaus Klingmüller[1], Vlassis A. Karydis[2], Sara Bacer[3], Georgiy L. Stenchikov[4], and Jos Lelieveld[1,5]

[1]Max Planck Institute for Chemistry, Hahn-Meitner-Weg 1, 55128 Mainz, Germany
[2]Forschungszentrum Jülich GmbH, IEK-8, 52425 Jülich, Germany
[3]LEGI, Université Grenoble Alpes, CNRS, Grenoble INP, Grenoble, France
[4]King Abdullah University of Science and Technology, Thuwal 23955-6900, Saudi Arabia
[5]The Cyprus Institute, P.O. Box 27456, 1645 Nicosia, Cyprus

**Correspondence:** Klaus Klingmüller (k.klingmueller@mpic.de)

**Abstract.** The interactions between aeolian dust and anthropogenic air pollution, notably chemical ageing of mineral dust and coagulation of dust and pollution particles, modify the atmospheric aerosol composition and burden. Since the aerosol particles can act as cloud condensation nuclei, this not only affects the radiative transfer directly via aerosol-radiation interactions, but also indirectly through cloud adjustments. We study both radiative effects using the global ECHAM/MESSy atmospheric chemistry-climate model (EMAC) which combines the Modular Earth Submodel System (MESSy) with the European Centre/Hamburg (ECHAM) climate model. Our simulations show that dust-pollution-cloud interactions reduce the condensed water path and hence the reflection of solar radiation. The associated climate warming outweighs the cooling that the dust-pollution interactions exert through the direct radiative effect. In total, this results in a net warming by dust-pollution interactions which moderates the negative global anthropogenic aerosol forcing at the top of the atmosphere by $(0.2 \pm 0.1)\,\mathrm{Wm^{-2}}$.

## 1 Introduction

A prime objective of current atmospheric and climate science is the deeper understanding of ambient aerosols and their interactions with clouds. This is motivated by their central role in two areas of societal importance, public health and climate change. The inhalation of aerosols allows fine particles to enter deep into the respiratory system or even translocate through the lungs into the cardiovascular system causing a multitude of health challenges, and making the exposure to fine particulate air pollution one of the main public health risks worldwide (Lelieveld et al., 2015, 2019a, b; Chowdhury et al., 2020). On the other hand, aerosols modify the albedo of the Earth, predominantly increasing the reflection of solar radiation and thus cooling the planet. Since the emissions of anthropogenic greenhouse gases are accompanied by those of aerosols, to a large extent through common source categories, the greenhouse warming has been partially masked by the aerosol effects on climate (IPCC, 2014; Lelieveld et al., 2019a).

The planetary albedo can be increased both directly by interactions of the anthropogenic aerosol particles with solar radiation and indirectly by enhanced cloudiness or cloud brightness caused by aerosol particles acting as cloud condensation nuclei. These direct and indirect effects are estimated to contribute a negative effective radiative forcing (ERF) of -0.45 $\mathrm{Wm^{-2}}$ each, adding up to about -0.9 $\mathrm{Wm^{-2}}$ (IPCC, 2014).

Since not all aerosols in the atmosphere are of anthropogenic origin – in fact natural aerosols including aeolian dust and sea salt are the most abundant components by mass – the anthropogenic pollutants form a mixture with natural aerosols. On the one hand particulate pollution coagulates with natural particles and on the other hand natural particles are exposed to chemical ageing. Klingmüller et al. (2019) showed that the interactions between natural mineral dust and anthropogenic pollution enhance the global net-cooling through the direct radiative effects and have a significant impact on regional radiative transfer. Here we extend the analysis to include the indirect radiative effects. The abundant atmospheric water vapour represents a vast source of cloud water so that cloud optical depths are typically much larger than aerosol optical depths. Therefore, cloud adjustments potentially leverage the aerosol radiative effect and we may expect the indirect radiative effect of the dust-pollution interactions to be even more significant than the direct effect.

We use the global ECHAM/MESSy atmospheric chemistry-climate model (EMAC) which combines the Modular Earth Submodel System (MESSy) with the European Centre/Hamburg (ECHAM) climate model. It includes implementations of an extensive set of relevant physical and chemical processes, including detailed parametrisations of mineral dust ageing, cloud droplet activation and ice crystal formation in cirrus and mixed-phase clouds.

The model and its configuration are described in Sect. 2 followed by an outline of the methodology of our analysis in Sect. 3. Results for the dust-pollution interaction effect on the cloud condensate are presented in Sect. 4, and the resulting effects on radiative transfer in Sect. 5. Conclusions are presented in Sect. 6.

## 2 Model description

The EMAC model version and configuration used in the present study are largely identical to those used by Klingmüller et al. (2018, 2019), combining ECHAM 5.3.02 and MESSy 2.52. However, to allow decadal simulations, the horizontal resolution has been reduced to a Gaussian T63 grid with a grid spacing of $1.875°$ along latitudes and about $1.86°$ along longitudes, corresponding to an edge length of the individual grid cells of around 200 km or less. The number of vertical levels remains at 31. Moreover, the present study uses the EDGARv4.3 (Emissions Database for Global Atmospheric Research) database for anthropogenic emissions (Crippa et al., 2016) and a backport of the CLOUD submodel from MESSy 2.54 to benefit from recent improvements of the cloud parametrisations.

As in the previous studies, the GFEDv3.1 (Global Fire Emissions Database) (Randerson et al., 2013) and AeroCom (Aerosol Comparisons between Observations and Models) (Dentener et al., 2006) databases provide biomass burning and sea salt emissions, respectively. Mineral dust emissions are calculated online by the submodel ONEMIS (Kerkweg et al., 2006b) using the dust emission scheme presented by Klingmüller et al. (2018) which is based on Astitha et al. (2012). It differentiates the $Ca^{++}$, $K^+$, $Mg^{++}$ and $Na^+$ fractions in mineral particles originating from different deserts (Karydis et al., 2016).

The MESSy submodels most relevant for aerosols include the Global Modal Aerosol Extension (GMXe) (Pringle et al., 2010a, b). It simulates the microphysics of four soluble (nucleation, Aitken, accumulation, coarse) and three insoluble (Aitken, accumulation, coarse) aerosol log-normal modes with fixed geometric standard deviations ($\sigma_g = 2$ for the coarse modes, $\sigma_g = 1.59$ for all others). The count median dry radius of each mode can vary between fixed boundaries at 6 nm, 60 nm and 1 μm.

Super coarse mineral dust particles are therefore only included as part of the coarse modes with mean radius larger than 1 μm and their mass is probably underrepresented (Adebiyi and Kok, 2020). However, their role in the dust-pollution-cloud interactions is limited by their low number concentration, corresponding to a low probability for pollution particles to coagulate with them, and a comparably short atmospheric residence time, leaving less time for chemical ageing.

Within GMXe, the gas-aerosol partitioning can be computed by ISORROPIA II (Fountoukis and Nenes, 2007) or EQSAM4clim (Equilibrium Simplified Aerosol Model V4 for climate simulations) (Metzger et al., 2016), here we use ISORROPIA II. Assuming diffusion limited condensation, it calculates the amount of gas kinetically able to condense using the accommodation coefficients in Table S1 in the supplement. Subsequently the mass is re-distributed between the gas and aerosol phase to obtain the amount of condensed material (Pringle et al., 2010a, b). This means that gaseous compounds from anthropogenic pollution,

including sulphuric acid, nitric acid, hydrochloric acid and ammonia, can condense on mineral dust particles and initiate their chemical ageing which is the primary interaction between mineral dust and gaseous pollution, primarily trough reactions of acids with mineral cations.

Insoluble particles are transferred to the soluble modes if sufficient hydrophylic material has accumulated to cover the particles with 10 molecular monolayers, or if they coagulate with soluble particles (Vignati et al., 2004; Stier et al., 2005;

Pringle et al., 2010a, b). In particular, freshly emitted mineral dust is assumed to be hydrophobic and thus emitted into the insoluble aerosol modes (with approximately 89 %, the majority of the mass is emitted into the coarse mode and the remainder into the accumulation mode), but chemical ageing can transfer the mineral dust particles to soluble modes. The chemical ageing and partitioning of organic aerosol compounds is implemented in the submodel ORACLE (Organic Aerosol Composition and Evolution) (Tsimpidi et al., 2014, 2018). GMXe and ORACLE interact with the gas phase for which the chemistry is simulated

by the submodel MECCA (Module Efficiently Calculating the Chemistry of the Atmosphere) (Sander et al., 2019).

In addition to the condensation of hydrophilic compounds, also the coagulation with soluble particles transfers insoluble particles to the soluble modes. Within GMXe, coagulation is implemented following Vignati et al. (2004) using the coagulation coefficient equation from Fuchs et al. (1965). All aerosol components are affected by coagulation irrespective of their sources and their chemical composition, including components represented by "bulk" tracers, which are treated as chemically

inert, and the major particulate pollutants black carbon, organic compounds, sulphates, nitrate and ammonium. This makes coagulation the primary interaction between mineral dust and particulate pollution. Aside from modifying the composition and hygroscopicity of dust particles, it has a significant effect on the burden of particulate pollution. Because typically mineral dust particles are coarser than pollution particles like soot or sulphate particles, coagulation with dust transfers fine particulate pollution to coarser modes, decreasing the number concentration especially in the fine modes. Once in the coarse mode, the

pollution is affected by the shorter atmospheric residence time of coarse particles, which reduces the mass concentration of particulate pollutants. After being transferred to the hydrophilic modes, mineral dust particles grow by taking up water and act as cloud condensation nuclei. The hygroscopic growth increases the deposition rate and affects the optical properties.

The AEROPT (AERosol OPTical properties) submodel (Lauer et al., 2007; Klingmüller et al., 2014) calculates the aerosol optical properties assuming the aerosol components within each mode to be well mixed in spherical particles with volume

averaged refractive index. The refractive indices considered by AEROPT for the individual components are compiled from the

OPAC 3.1 database (Hess et al., 1998) (black carbon, mineral dust), the HITRAN 2004 database (Rothman et al., 2005) (organic carbon, sea salt, ammonium sulphate, water), Kirchstetter et al. (2004) (organic carbon for $\lambda < 0.7\mu$m) and additional mineral dust values for $\lambda > 2.5\mu$m. The full dataset is specified in the supplement of Klingmüller et al. (2014). The imaginary part of the dust refractive index provided by the OPAC dataset attains a minimum of $4 \cdot 10^{-3}$ at visible and near-infrared wavelengths.

This is lower than the former recommendation by the World Metorological Organisation of $8 \cdot 10^{-3}$ (Deepak et al., 1983), but even smaller and regionally varying values are found in more recent literature (Kaufman et al., 2001; Müller et al., 2011; Di Biagio et al., 2019). Even though a larger imaginary part of the refractive index corresponds to stronger absorption, we obtain a distinctive negative climate forcing attributed to mineral dust (Tab. 2). In our simulations the modelled dust is usually internally mixed with other components and especially water so that the effective imaginary refractive index of the entire

particles is often lower than the value assumed for pure dust. Using a smaller value for pure dust would further enhance the negative direct forcing of dust and to some extent the direct forcing through the dust-pollution interactions. However, the dominant indirect effect of the interactions would not be affected.

    The aerosol optical properties are considered by the radiative transfer submodel RAD (Dietmüller et al., 2016) to account for the aerosol-radiation coupling. In the solar spectrum, absorption and scattering are computed using extinction coefficient, single

scattering albedo and asymmetry parameter, whereas in the terrestrial spectrum scattering is neglected. The latter approximation is valid for particles much smaller than the wavelength and is therefore largely justified for the long terrestrial wavelengths, but might be inaccurate in the presence of super coarse particles (Di Biagio et al., 2020). However, this only affects the direct radiative effect. In the context of the present study, the indirect radiative effect turns out to be much more relevant.

    Aerosol removal by wet deposition is calculated by the scavenging submodel SCAV (Tost et al., 2006a), dry deposition

and sedimentation by the submodels DDEP and SEDI (Kerkweg et al., 2006a). The aerosol and in particular the mineral dust representation in EMAC have a proven track record (e.g., Abdelkader et al., 2015; Metzger et al., 2016; Abdelkader et al., 2017; Klingmüller et al., 2018; Brühl et al., 2018; Metzger et al., 2018; Ma et al., 2019), the dust aerosol optical depth is consistent with observations not only at visible wavelengths but also in the infrared at 10 µm (Klingmüller et al., 2018). This is an indication of a realistic particle size distribution, provided that the ratio of the extinction efficiencies at visible and infrared

wavelengths is realistic. The uncertainty in this ratio is expected to be small compared to other uncertainties, given that the spectral extinction efficiency is calculated consistently throughout the spectrum and, unlike the single scattering albedo, is hardly sensitive to the aforementioned uncertainties in the imaginary part of the refractive index.

    Large-scale clouds are simulated by the submodel CLOUD (Jöckel et al., 2006) where different parametrisations of cloud droplet formation and ice nucleation are implemented. We use a two-moment stratiform cloud microphysics scheme (Lohmann

et al., 1999, 2007; Lohmann and Kärcher, 2002) in combination with the UAF (Unified Activation Framework) cloud droplet activation parametrisation (Kumar et al., 2011; Karydis et al., 2011, 2017). For the ice crystal formation we use the comprehensive parametrisation for cirrus and mixed-phase clouds implemented by Bacer et al. (2018) based on Barahona and Nenes (2009). Convective clouds are calculated by the CONVECT submodel (Jöckel et al., 2006), where interactions with aerosols are not taken into account. CONVECT provides a choice of convection schemes (Tost et al., 2006b), and here we use the

scheme of Tiedtke (1989) including modifications by Nordeng (1994). The optical properties of clouds which serve as input

for the radiative transfer submodel RAD are computed by the submodel CLOUDOPT (Dietmüller et al., 2016). The model yields a global annual mean cloud liquid water path around $80 \, \mathrm{g\,m^{-2}}$ (Table 1 and Table S3 in the supplement), which is well within the range of other climate model results ($32$ to $125 \, \mathrm{g\,m^{-2}}$, Lebsock and Su, 2014) and observations ($30$ to $90 \, \mathrm{g\,m^{-2}}$, Lohmann and Neubauer, 2018). Likewise, the modelled annual mean global cloud ice water path of about $15 \, \mathrm{g\,m^{-2}}$ (Table 1 and Table S3 in the supplement) is consistent with results from other models (e.g., $14.8 \, \mathrm{g\,m^{-2}}$, Lohmann and Neubauer, 2018) and close to observed values (e.g., $(25 \pm 7) \, \mathrm{g\,m^{-2}}$, Li et al., 2012).

A complete list of the MESSy submodels used in our simulations is provided in Table S2 in the supplement. Descriptions of each submodel and further references can be found online in the MESSy submodel list (MESSy, 2020).

## 3  Methodology

We apply a similar analysis as Klingmüller et al. (2019) which is based on simulations with four different emission set-ups: a baseline simulation with neither dust nor anthropogenic emissions ("0"), a simulation with dust but without anthropogenic emissions ("dust"), a simulation with anthropogenic pollution but without dust emissions ("pol") and a full simulation considering all emissions.

In the anthropogenic pollution free simulations ("0", "dust") we disable the EDGAR emissions including $SO_2$, $NH_3$, $NO_x$, black- and organic carbon emissions, but retain the greenhouse gases. We attribute 90 % of the GFED biomass burning emissions to human activities (Levine, 2014) and reduce them accordingly, whereas we do not consider anthropogenic factors on dust emissions such as land use and climate change (Klingmüller et al., 2016), assuming all dust emissions to be natural.

A result $x$ from the full simulation (e.g., the annual global mean cloud liquid water content) is related to the corresponding result from the baseline simulation $x_0$ by

$$x = x_0 + \Delta_{\mathrm{dust}}x + \Delta_{\mathrm{pol}}x + \Delta_{\mathrm{int}}x \tag{1}$$

where $\Delta_{\mathrm{dust}}x = x_{\mathrm{dust}} - x_0$, $\Delta_{\mathrm{pol}}x = x_{\mathrm{pol}} - x_0$ and

$$\Delta_{\mathrm{int}}x = x - x_{\mathrm{dust}} - x_{\mathrm{pol}} + x_0, \tag{2}$$

which represents the effect of the dust-pollution interactions. In the absence of such interactions, the term $\Delta_{\mathrm{int}}x$ vanishes. We apply Eq. (2) to the annual mean cloud liquid- and ice-water paths and radiative fluxes.

To quantify the effects of the different emission set-ups and the dust-pollution interactions on radiation, we consider the effective radiative forcing (ERF) which is defined as the change in net TOA downward radiative flux after allowing for atmospheric temperatures, water vapour and clouds to adjust, but with sea surface temperatures (SST) and sea ice cover fixed at climatological values (IPCC, 2014). Note that positive downward fluxes correspond to downward (incoming) radiation, negative values correspond to upward (outgoing) radiation. The ERF accounts for rapid adjustments by radiative and dynamical feedbacks, whereas it excludes long-term climate responses involving the much slower thermal equilibration of the oceans. Due to the limited constraints on the atmospheric dynamics in SST simulations, the meteorological variability is large and hence a sufficient number of years has to be simulated to obtain statistically significant results. We perform SST simulations

long enough to yield significant globally averaged results, however detailed regional analysis would require much longer SST simulations. In order to nevertheless gain insights from regional evaluation, we additionally use simulations where the model dynamics above the boundary layer is nudged to meteorological analyses of the European Centre for Medium-Range Weather Forecasts (ECMWF). Within the boundary layer, in the topmost layers and to some extend in-between, nudged quantities like the temperature may still respond to other variables such as radiative fluxes (soft nudging). The nudging greatly reduces the influence of inter-annual variability on statistical analysis. The results from the nudged simulations turn out to be largely consistent with those of the SST simulations (Tabs. 1, 2 vs. Tabs. S3, S4 in the supplement), in particular the estimates for the total global radiative effect of the dust-pollution interactions agree within the error bounds, so that the use of nudged simulations for the regional analysis is reasonable and helpful.

With prescribed SST we run ensembles of 16 simulations, each covering one year. As there is one ensemble for each of the four emission set-ups, in total this amounts to 64 SST simulations. The ensemble members are obtained by perturbing temperature and humidity in the fourth year of a common spin-up simulation, followed by an additional spin-up of the individual ensemble members to attain a total of 5 spin-up years. The perturbation is implemented by adding a uniformly distributed random variable ranging from -0.1 ‰ to 0.1 ‰ of the perturbed quantity so that the perturbation is numerically but not meteorologically relevant. Emission data for 2010 is used for all simulations. The nudged simulations cover 10 years from 2006 to 2015, and two simulation years prior to that period were used for the model spin-up. To estimate the uncertainties of the 10-year mean values for the nudged simulations and the ensemble mean values for the SST simulations, we compute the standard error of the mean (SEM) of the annual values.

In the analysis of the SST results, we substitute the variable $x$ in Eq. (2) with global annual mean values, and for the nudged results we skip the global averaging and apply the equation to the annual mean for each grid cell separately to obtain the spatial distribution of the interaction term.

Substituting $x$ for the global annual mean net-flux $F$ at the top of the atmosphere (TOA) in the SST simulations, $\Delta_{\mathrm{dust}}F = F_{\mathrm{dust}} - F_0$ corresponds to the total ERF of mineral dust including all rapid adjustments, analogously $\Delta_{\mathrm{pol}}F$ to the anthropogenic aerosol ERF (both excluding the dust-pollution interactions) and $\Delta_{\mathrm{int}}F$ to the ERF of dust-pollution interactions. In case of the nudged simulations, the possible adjustments are constrained so that so that the resulting forcings are in-between the ERF and the radiative forcing RF as defined by IPCC (2014) where only the stratospheric temperature is allowed to adjust. For this reason the forcings from the nudged simulations are not directly comparable to RF and ERF results, but as mentioned above, provide valuable information about the regional effects.

To compute the direct radiative effect of aerosols, the radiative transfer code is called twice for every model time step. The first call considers scattering and absorption by aerosols and is used to calculate the heating rates affecting the temperature, the second call ignores scattering and absorption by aerosols and computes the radiative fluxes and heating rates only for diagnostic output. The difference of the radiative fluxes from both calls yields the instantaneous forcing (IRF) due to the direct radiative effect of aerosols $F_{\mathrm{ari}}$. Since both calls are performed with identical clouds, the cloud forcing is excluded and only little statistical noise is introduced by the strong variability of clouds. Nevertheless, in this way we obtain the direct radiative forcing in the presence of clouds, which is typically smaller than the clear sky forcing. The difference of the instantaneous

aerosol forcings in the SST simulations with and without mineral dust $\Delta_{dust}F_{ari} = F_{ari,dust} - F_{ari,0}$ yields the aerosol-radiation interaction contribution to the ERF of dust, i.e., the direct radiative effect of dust. Analogously $\Delta_{pol}F_{ari}$ and $\Delta_{int}F_{ari}$ represent the direct radiative effect of particulate pollution and the dust-pollution interactions. The direct radiative forcings are subtracted from the corresponding total aerosol radiative forcings to extract the indirect radiative forcings, e.g., the indirect contribution
to the dust-pollution interaction forcing is $\Delta_{int}F - \Delta_{int}F_{ari}$.

## 4   Effects on the cloud condensate

Hydrophilic particulate anthropogenic pollution enhances the cloud droplet formation and thus the liquid water content (Table 1). However, in the presence of mineral dust particles this effect is reduced because fine pollution particles coagulate with coarse dust particles decreasing the particle number and virtually cleaning the atmosphere from fine particulate pollution.
Moreover, the adsorption activation of mineral dust particles occurs early on in the cloud formation process (Kumar et al., 2011), reducing the maximum supersaturation and inhibiting the activation of small pollution particles. These effects reduce the number of cloud condensation nuclei (Karydis et al., 2011, 2017) and decrease the cloud liquid water path as shown in Fig. 1 (a). Especially over East and South Asia, where strong pollution emissions mix with aeolian dust from the Taklamakan, Gobi and Thar deserts, the reduction is substantial and regionally exceeds -40 $\mathrm{g\,m^{-2}}$. Even over polluted regions in Europe
and the USA which are only occasionally exposed to dust intrusions, we obtain a small but significant reduction. This negative impact of the dust-pollution interactions over large parts of the northern hemisphere leads to a reduction of the global mean cloud liquid water path in Fig. 1 (a) by (-1.10 ± 0.03) $\mathrm{g\,m^{-2}}$. A comparable reduction by (-1.5 ± 0.2) $\mathrm{g\,m^{-2}}$ is obtained in the SST simulations (Tab. 1). Relative to the mean liquid water path in the SST simulation considering all emissions (85.5 ± 0.1) $\mathrm{g\,m^{-2}}$, these reductions appear to be rather moderate. The reason is that the transport time periods between most of
the major dust sources, especially the Sahara and the Middle East, and major pollution sources like Northern America and Europe to a large degree exceed the dust aerosol lifetime. In Asia these sources are less distant while pollution emissions are generally larger. Thus, the strong effects over Asia might provide an outlook for regions with emerging pollution sources close to dust sources in Africa and the Middle East. But already today, due to the critical influence of clouds on radiative transfer, the relatively small changes of the water paths cause substantial radiative forcings as will be discussed in the next section.
The dust-pollution interaction effect on the cloud ice water path, shown in Fig. 1 (b), is less distinct. A negative impact is obtained over the Sahel. The direct radiative effect of mineral dust over the Sahara warms the atmosphere by absorption of solar radiation (Fig. S1 (a) in the supplement). This increases the atmospheric capacity to hold moisture and the vertical water vapour transport (Fig. S1 (b) in the supplement). As a result, more moisture is available for ice cloud formation (Fig. S1 (c) in the supplement). Since the net direct radiative effect of the dust-pollution interactions cools the atmosphere over the Sahara
(Klingmüller et al., 2019), it moderates the enhancement of ice cloud formation. A similar net cooling effect is found over the region around the Taklamakan and Gobi deserts. In this region with generally high ice water content, anthropogenic pollution enhances the ice water path, but adding dust reduces the number of anthropogenic ice nucleation particles via coagulation. In contrast, a positive impact is obtained over coastal regions of Canada and Greenland around 60 degrees north, probably due

to aerosol and cloud feedbacks on the polar and Ferrel cell circulations and associated vertical moisture transport. However, because of the comparably small radiative fluxes at these latitudes, this has relatively little impact on radiative fluxes from a global perspective. Due to the regionally varying sign of the dust-pollution interaction effect on cloud ice, the global mean in Fig. 1 (b) is close to zero, $(-0.027 \pm 0.003)\,\mathrm{g\,m^{-2}}$. The corresponding value in the SST simulations, $(-0.02 \pm 0.03)\,\mathrm{g\,m^{-2}}$, is consistent with this result, being several orders of magnitude smaller than the global mean ice water path in the SST simulation considering all emissions, $(14.70 \pm 0.01)\,\mathrm{g\,m^{-2}}$ (Tab. 1).

## 5  Radiative effects

The reduction of the cloud water content by dust-pollution interactions has a significant impact on the transfer of solar radiation ("shortwave", SW), which is shown in Fig. 2 (a). With reduced liquid cloud water, less solar radiation is reflected back to space, i.e., the outgoing radiation and the associated negative contribution to the net flux decreases, corresponding to a net positive forcing at the TOA. Comparing Fig. 2 (a) and Fig. 1 (a) reveals the one-to-one correspondence of the dust-pollution interaction effect on the liquid cloud water and solar radiation. Over the polluted regions of the northern hemisphere, i.e., Asia, Europe and North America, and over the Atlantic Ocean along the North African coast in the Saharan dust outflow, the positive forcing can exceed 2 $\mathrm{Wm^{-2}}$. Globally averaged, the net forcing in the solar spectrum shown in Fig. 2 (a) is $(0.23 \pm 0.01)\,\mathrm{Wm^{-2}}$, the SST simulations yield an ERF of $(0.3 \pm 0.1)\,\mathrm{Wm^{-2}}$ (Tab. 2).

On the other hand, the dust-pollution interaction effect on the terrestrial spectrum ("longwave", LW), Fig. 2 (b), is directly related to the effects on ice clouds, Fig. 1 (b). This is most distinct over the Sahel, but also apparent over the East Asian deserts. The reduced cloud ice water path over these regions traps less outgoing terrestrial radiation resulting in a net cooling from the dust-pollution interactions. Over the Sahel the terrestrial TOA forcing reaches -2 $\mathrm{Wm^{-2}}$. With regard to the radiative energy budget, the regions with a significant dust-pollution interaction effect on cloud ice in Fig. 1 (b) are of different relevance. The Sahel, where the dust-pollution interactions reduce cloud ice, is relatively close to the equator and accordingly stronger radiative fluxes are affected by the cloud ice changes than in the other regions, hence the global net radiative effect related to cloud ice is more relevant than the global net effect on cloud ice itself. Globally averaged, the net forcing in the terrestrial spectrum shown in Fig. 2 (b) is $(-0.05 \pm 0.01)\,\mathrm{Wm^{-2}}$, and the SST simulations yield an ERF of $(-0.08 \pm 0.09)\,\mathrm{Wm^{-2}}$ (Tab. 2).

Thus, a substantial positive forcing in the solar spectrum is partially compensated by a negative forcing in the terrestrial spectrum to yield a still considerable, positive net-forcing associated with the effect of dust-pollution interactions on clouds. The global distribution of the total net-forcing at the TOA including the direct radiative effect is shown in Fig. 3. The regional forcing ranges from below -2 $\mathrm{Wm^{-2}}$ over the Sahel to above 2 $\mathrm{Wm^{-2}}$ over Asia. Even though overall these contributions partially counterbalance, with $(0.15 \pm 0.02)\,\mathrm{Wm^{-2}}$ the corresponding global mean forcing in Fig. 3 is significantly positive. Consistently, the ERF in the SST simulations is $(0.2 \pm 0.1)\,\mathrm{Wm^{-2}}$ (Tab. 2).

Figure 4 summarises the direct and indirect global TOA ERF of anthropogenic aerosol interacting with mineral dust and in the absence of mineral dust, obtained from the SST simulations. Despite the more negative anthropogenic aerosol direct radiative forcing in the presence of mineral dust, already reported by Klingmüller et al. (2019), the effect of mineral dust on

the total forcing is clearly dominated by the moderation of the indirect forcing. The figure highlights the importance of the dust-pollution interactions for assessing the cooling effect of anthropogenic aerosol: the cooling is substantially reduced by the interactions from (-0.81 ± 0.06) Wm$^{-2}$ ERF, which is close to 0.9 Wm$^{-2}$ estimated by IPCC (2014), down to (-0.6 ± 0.1) Wm$^{-2}$.

## 6   Conclusions

We have studied the effects of interactions between mineral dust and anthropogenic pollution on clouds and radiation by analysing comprehensive global simulations performed with the atmospheric chemistry-climate model EMAC. Four different emission configurations representing all possible combinations of in- and excluding dust and pollution were considered. Comparing the results for these four scenarios allowed us to isolate the effect of the dust-pollution interactions from the individual effects of dust and pollution. Several aspects make this analysis challenging and should be considered when interpreting the results, and may leave room for refinements in future studies. Naturally, clouds are subject to strong variability, hence although we performed ensemble simulations there is a considerable statistical uncertainty in the present results. This adds to the need to evaluate differences in differences of results from a number of simulations to obtain the interaction effect, which increases the relative error. Moreover, a wide range of physical and chemical processes is involved in the dust-pollution interactions, and accordingly many submodels and parametrisations within EMAC contribute to our final result and uncertainty. Even though the parametrisations are well established and tested, the analysis might be sensitive to systematic errors of some of them.

The analysis reveals that the cloud water path is reduced by the dust-pollution interactions as they moderate the cloud water path increase caused by anthropogenic pollution. The reason for this moderation is that mineral dust particles decrease the number of anthropogenic cloud condensation nuclei by coagulation and additionally limit the activation of the fine hydrophilic anthropogenic particles by lowering the maximum supersaturation through adsorption activation. Dust-pollution interaction effects on the cloud ice content are noticeable as well, but less relevant.

The atmospheric radiative transfer is very sensitive to the reduction of the cloud water path. Generally, dust-pollution interactions affect the radiative transfer at all wavelengths (solar and terrestrial) by modifying both the direct aerosol-radiation interactions and the indirect radiative effect of aerosols via cloud adjustments. However, the total radiative effect of the dust-pollution interactions is dominated by the impact through the indirect effect which, in contrast to the direct effect, exerts an overall positive TOA net-forcing. The impact on the indirect radiative effect in turn is dominated by that on solar radiation fluxes. In this case, the aforementioned decrease of the cloud water path reduces the cloud albedo and the reflection of solar radiation, resulting in a positive contribution to the radiative net-flux.

We estimate that dust-pollution interactions contribute (0.2 ± 0.1) Wm$^{-2}$ to the global mean anthropogenic aerosol effective radiative forcing, significantly reducing the climate cooling effect of atmospheric aerosols. In view of this considerable contribution to the atmospheric energy balance, it is recommended to account for the dust-pollution interactions in assessments of climate change especially because on a regional scale effects can be even larger. The net global effect partially depends on regionally counteracting positive and negative radiative forcings. This study emphasizes the importance of continued efforts to

improve the understanding and parametrisations of the processes involved in order to reduce the uncertainty of future climate simulations.

*Code and data availability.* The Modular Earth Submodel System (MESSy) is continuously further developed and applied by a consortium of institutions. The usage of MESSy and access to the source code is licensed to all affiliates of institutions which are members of the MESSy Consortium. Institutions can become a member of the MESSy Consortium by signing the MESSy Memorandum of Understanding. More information can be found on the MESSy Consortium Website (https://www.messy-interface.org, last access: 30 October 2020). The ECHAM climate model is available to the scientific community under the MPI-M Software License Agreement (https://mpimet.mpg.de/en/science/modeling-with-icon/code-availability, last access: 30 October 2020). The simulation results analysed in this study are available at https://edmond.mpdl.mpg.de/imeji/collection/V5fqhlhJgMJAJ3 (Klingmüller, 2020).

*Author contributions.* KK performed the simulations assisted by VAK and SB, analysed the model results and wrote the article supported by JL, SB, VAK and GLS. All authors discussed the results and contributed to the final manuscript.

*Competing interests.* The authors declare that they have no conflict of interest.

*Acknowledgements.* The research reported in this publication has received funding from the MaxWater initiative of the Max Planck Society and the King Abdullah University of Science and Technology (KAUST) CRG3 grant URF/1/2180-01-01 "Combined Radiative and Air Quality Effects of Anthropogenic Air Pollution and Dust over the Arabian Peninsula".

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

# Dust-pollution interaction effect on clouds

## (a) Liquid water path

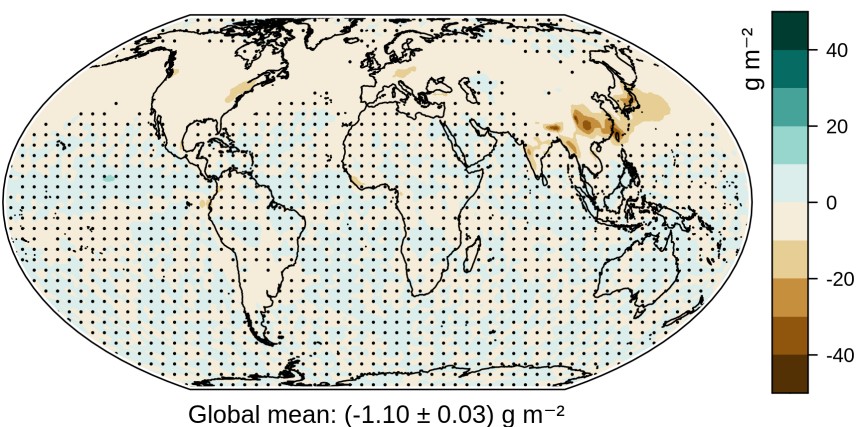

Global mean: (-1.10 ± 0.03) g m⁻²

## (b) Ice water path

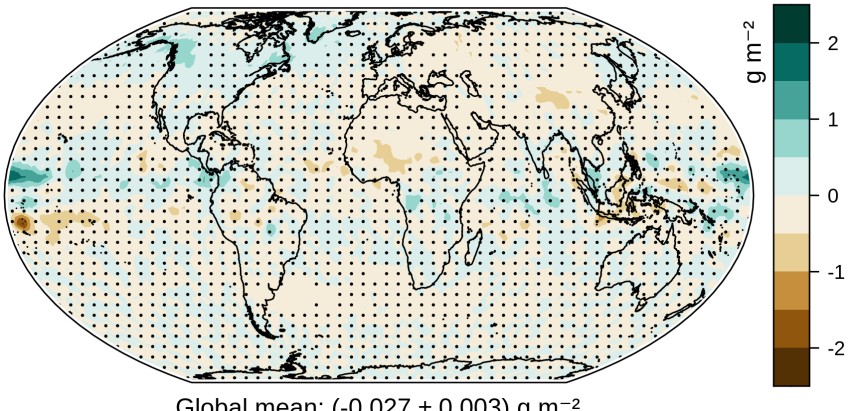

Global mean: (-0.027 ± 0.003) g m⁻²

**Figure 1.** Annual mean effect of the dust-pollution interactions on the liquid (a) and ice (b) cloud water, calculated by applying Eq. (2) to the results of the nudged simulations. Over stippled regions the results differ from zero by less than two times the SEM of the annual values.

# Dust-pollution interaction effect on the indirect aerosol TOA forcing

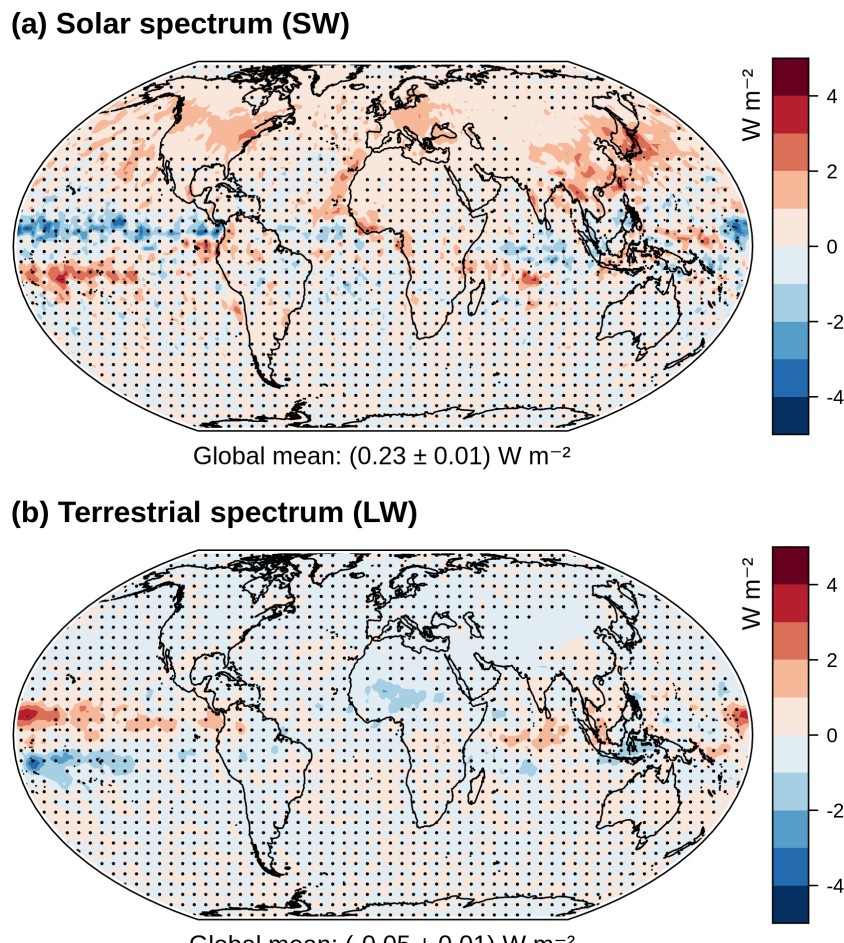

## (a) Solar spectrum (SW)

Global mean: (0.23 ± 0.01) W m⁻²

## (b) Terrestrial spectrum (LW)

Global mean: (-0.05 ± 0.01) W m⁻²

**Figure 2.** Annual mean indirect effect of the dust-pollution interactions on the solar (a) and terrestrial (b) radiative forcing at the top of the atmosphere, calculated by applying Eq. (2) to the results of the nudged simulations. Over stippled regions the results differ from zero by less than two times the SEM of the annual values.

## Dust-pollution interaction effect on the aerosol TOA forcing

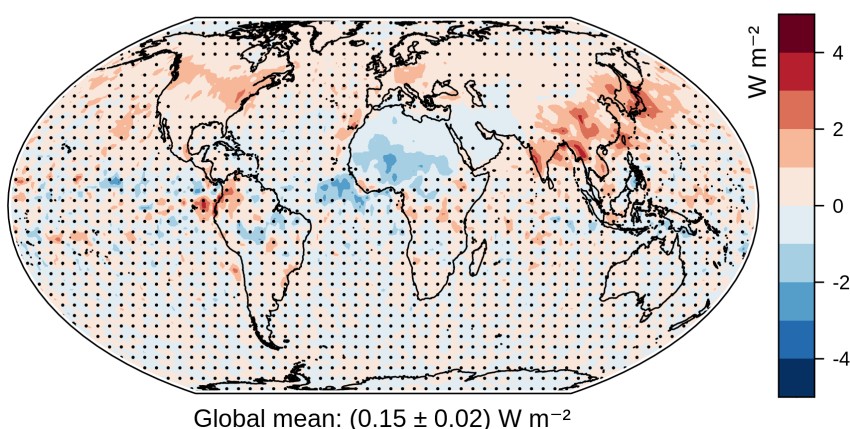

Global mean: (0.15 ± 0.02) W m$^{-2}$

**Figure 3.** Total (direct and indirect, SW and LW) annual mean effect of the dust-pollution interactions on the radiative forcing at the top of the atmosphere, calculated by applying Eq. (2) to the results of the nudged simulations. Over stippled regions the results differ from zero by less than two times the SEM of the annual values.

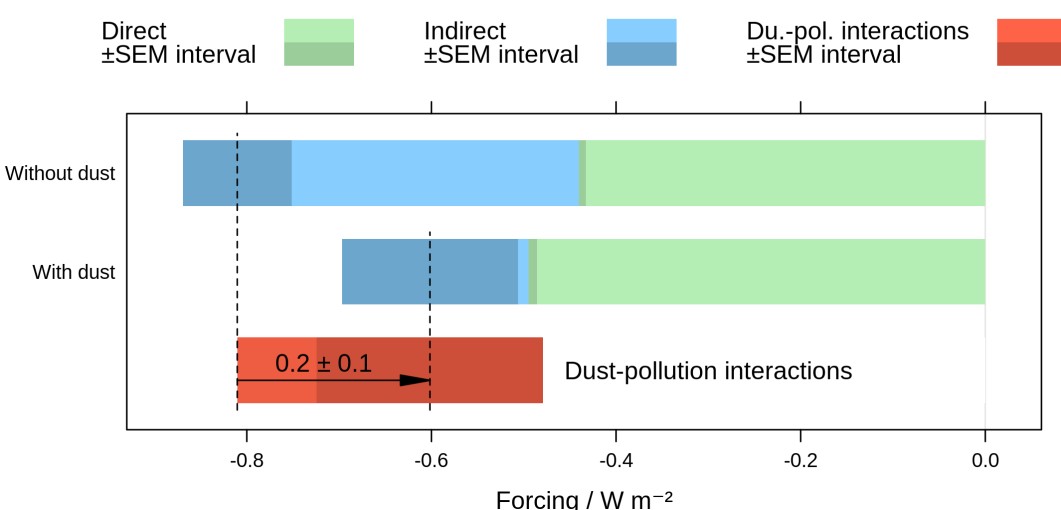

**Figure 4.** Estimates of the global anthropogenic aerosol forcings at the top of the atmosphere (TOA) in the presence ("With dust", $F - F_{\text{dust}}$) or absence ("Without dust", $F_{\text{pol}} - F_0$), of aeolian dust, based on the SST simulations. The total forcings comprise the direct (green, $F_{\text{ari}} - F_{\text{ari,dust}}$ and $F_{\text{ari,pol}} - F_{\text{ari,0}}$) and indirect (blue) forcings. The change caused by including mineral dust corresponds to the positive forcing of dust-pollution interactions $\Delta_{\text{int}} F$ (red). Darker colours represent the standard error of mean (SEM).

**Table 1.** Globally averaged annual mean cloud properties and contributions thereto, based on the SST simulations. "Total" represents the simulation with all emissions, "Mineral dust" and "Anthropogenic pollution" include the effect of dust-pollution interactions ($\Delta_{\mathrm{dust}}x + \Delta_{\mathrm{int}}x$ and $\Delta_{\mathrm{pol}}x + \Delta_{\mathrm{int}}x$ in Eq. (1)), "Dust-pollution interactions" are given by the interaction term $\Delta_{\mathrm{int}}x$, Eq. (2). The corresponding results from the nudged simulations are provided in Table S4 in the supplement.

|  | Total | Mineral dust | Anthropogenic pollution | Dust-pollution interactions |
|---|---|---|---|---|
| Droplet number / $\mathrm{m}^{-2}$ | $(5.845 \pm 0.009) \times 10^{10}$ | $(-2.2 \pm 0.1) \times 10^{9}$ | $(5.1 \pm 0.1) \times 10^{9}$ | $(-2.2 \pm 0.1) \times 10^{9}$ |
| Liquid water path / $(\mathrm{g\ m}^{-2})$ | $85.5 \pm 0.1$ | $-1.5 \pm 0.1$ | $1.6 \pm 0.2$ | $-1.5 \pm 0.2$ |
| Ice water path / $(\mathrm{g\ m}^{-2})$ | $14.70 \pm 0.01$ | $-0.04 \pm 0.02$ | $0.60 \pm 0.02$ | $-0.02 \pm 0.03$ |

**Table 2.** Globally averaged annual mean TOA ERFs in W m$^{-2}$, based on the SST simulations. "Mineral dust" and "Anthropogenic pollution" include the effect of dust-pollution interactions ($\Delta_{\mathrm{dust}}x + \Delta_{\mathrm{int}}x$ and $\Delta_{\mathrm{pol}}x + \Delta_{\mathrm{int}}x$ in Eq. (1)), "Dust-pollution interactions" are given by the interaction term $\Delta_{\mathrm{int}}x$, Eq. (2). The corresponding forcings obtained from the nudged simulations are provided in Table S4 in the supplement.

| | | Mineral dust | Anthropogenic pollution | Dust-pollution interactions |
|---|---|---|---|---|
| | Total | -0.01 ± 0.07 | -0.6 ± 0.1 | 0.2 ± 0.1 |
| Net | Direct | -0.260 ± 0.006 | -0.490 ± 0.005 | -0.054 ± 0.005 |
| | Indirect | 0.25 ± 0.07 | -0.1 ± 0.1 | 0.3 ± 0.1 |
| | Total | -0.04 ± 0.06 | -0.86 ± 0.06 | 0.3 ± 0.1 |
| SW | Direct | -0.367 ± 0.007 | -0.527 ± 0.005 | -0.058 ± 0.005 |
| | Indirect | 0.33 ± 0.07 | -0.33 ± 0.06 | 0.3 ± 0.1 |
| | Total | 0.03 ± 0.08 | 0.26 ± 0.07 | -0.07 ± 0.09 |
| LW | Direct | 0.107 ± 0.001 | 0.0368 ± 0.0009 | 0.004 ± 0.001 |
| | Indirect | -0.08 ± 0.08 | 0.22 ± 0.07 | -0.08 ± 0.09 |