# Peer review of "Weaker cooling by aerosols due to dust-pollution interactions"

_Atmospheric Chemistry and Physics, 2020_

## Referee Comment (RC1) · Anonymous Referee #1 · 14 Jul 2020

The study described in the manuscript uses the atmospheric chemistry model EMAC to evaluate the direct and indirect radiative effects of dust-pollution interactions, i.e. as mineral dust particles are allowed to act as a surface for condensation of aerosol precursors, with the effect of changing the aerosol mix optical properties, hygroscopicity, number, and size. I found the manuscript and its results interesting, and generally well written and clear. I recommend that some aspects are clarified before publication.

p. 3, 19 – Could you give a few more details on what are the actual dust-pollution interactions happening in the model? i.e. which are the other aerosol species interacting with dust? Can you specify what "pollution" is exactly here? Black carbon?

p. 3, 24 - Please discuss dust optical properties in your model in light of the relevant literature (e.g. Kaufman et al. 2001, Müller et al., 2011, Di Biagio et al. 2019). It is

relevant to assess this potential source of uncertainty.

p. 3, 34 - The indirect statement about the particle size distributions seems unjustified. AOD at a given wavelength is usually parameterized as the product of aerosol column loading for a given size mode/bin times the size-dependent mass extinction efficiency. At least, the statement should be supported by additional evidence that the optical properties that you use are "reliable".

p. 5, 11 – Is there a spatial component in your procedure of reducing anthropogenic biomass burning emissions?

p. 5, 20 – Can you elaborate on the two different strategies for averaging?

p. 5, 25 – Please clarify what your strategy is, in light of e.g. IPCC AR5 terminology: "The difference of the radiative fluxes from both calls yields the instantaneous forcing . . ." vs e.g. "Globally averaged, the net forcing in the solar spectrum shown in Fig. 2 (a) is (0.23 ± 0.01) Wm−2, the SST simulations yield an ERF of (0.3 ± 0.1) Wm−2" (p. 7, 8). Please clarify how each metric that you will discuss is calculated, and try to be consistent in the terminology you chose, in differentiating between instantaneous RF, ERF, and direct and indirect radiative effects.

p. 6, 6 – In light of your strategy (i.e. experimental design) is that because of a local aerosol effect or else?

p. 8, 17 – "Our SST simulations . . ." please rephrase, and resume briefly your experimental setup in the conclusions. Also, in this section you should discuss the limitations related to the experimental setup, as suggested by the editor.

Figures 1, 2, 3 - It would be useful to see the individual (dust, pollution) effects alongside with their interactions. It would also be useful to see the average dust and pollution burden maps somewhere also within this manuscript.

Figure captions – "Over stippled regions the results are consistent with zero at 2 $\sigma$ significance level." What do you mean by "zero"? A null hypothesis of zero difference

in the mean among the ensemble members? Please clarify the captions, and provide the details of your procedure.

Figure 4 – The bar chart is not very clear in its present form. Please make it explicit which simulation ensembles you are depicting in each of the top bars: only pollution and dust+pollution?

References

Kaufman, Y. J., D. Tanré, O. Dubovik, A. Karnieli, and L. A. Remer, 2001: Absorption of sunlight by dust as inferred from satellite and ground-based remote sensing. Geophys. Res. Lett., 28, 1479–1482, https://doi.org/10.1029/2000GL012647.

Müller, T., A. Schladitz, K. Kandler, and A. Wiedensohler, 2011: Spectral particle absorption coefficients, single scattering albedos and imaginary parts of refractive indices from ground based in situ measurements at Cape Verde Island during SAMUM-2. Tellus, 63B, 573–588, https://doi.org/10.1111/ j.1600-0889.2011.00572.x.

Di Biagio, C., Formenti, P., Balkanski, Y., Caponi, L., Cazaunau, M., Pangui, E., et al. (2019). Complex refractive indices and single scattering albedo of global dust aerosols in the shortwave spectrum and relationship to iron content and size. Atmospheric Chemistry and Physics Discussions, 19(24), 15,503–15,531. https://doi.org/10.5194/acp‐2019‐145.

---

## Referee Comment (RC2) · Anonymous Referee #2 · 6 Aug 2020

This manuscript describes global model simulations to assess the impact of dust pollution interactions in cloud liquid and ice water content and direct and indirect aerosol radiative effect. The design of numerical simulations is appropriate and methodology to delineate various terms is reasonable. The results indicate dust aerosol interaction will reduce the negative cooling effect of aerosol. The result is quite interesting and worth publication. However, some clarifications of the methodology are required to facilitate better understanding.

P3, L28-29, The sentence reads awkward.

The methodology section, it would be better to move the description of four experiments with ('0', 'dust','Anthropogenic','Full') first. Then describe the 16 ensemble simulations and the nudged simulations for each experiment.

[Figure]

Table 1-2 and 3S-4S caption, I am very confused here. The Mineral dust and Anthropogenic pollution by their definition, either only contains dust or pollution and there should be no dust pollution interaction, why do these simulations include the interactive term? Shouldn't 'Mineral dust' effect be simply Xdust – X0 and "Anthropogenic" effect be Xpol – X0 and the last term computed with Eq. 2?

P5 L23-24. What do you mean by neglecting 'aerosol-radiation-interaction'? Do you mean by excluding the aerosol contribution in the radiation calculation? That should not be termed as "aerosol-radiation-interaction".

P5 L28. How do you compute the total radiative forcing? Do you mean "total aerosol radiative forcing"?

Section 5. How do you define the TOA radiative forcing in SW and LW? Is this simply the difference of reflective SW and outgoing long wave (LW)? What direction is considered positive, into the Earth or out of Earth?

P7 Last paragraph and Figure 4: it contains a lot of calculations that are not straitforward to readers. It took me a while to figure out (hopefully I got them correct!). Better to spell out how each term is calculated. For example, Xp – X0 represents total aerosol effect without dust and X – X0 represent total aerosol effect with dust (blue + green). The green part is computed from the difference of two calls of radiative transfer code with or without aerosol contributions. Then the blue part is total minus the green part. The red bar should be result of Eq. (2). Why is this number different from the global total in Figure 3?

I haven't understood the rational for using maps from nudged simulations while global averaged effect (Table 1 and 2) from SST simulations in the main article. Could you explain how each of these simulation configurations contrast and complement with the story you which to tell?

---

## Author Comment (AC1) · 30 Sep 2020

**Reply to RC1**

We thank the reviewer for the constructive and encouraging review which was very valuable for improving the manuscript. Below please find the point by point reply to the comments.

> The study described in the manuscript uses the atmospheric chemistry model EMAC to evaluate the direct and indirect radiative effects of dust-pollution interactions, i.e. as mineral dust particles are allowed to act as a surface for condensation of aerosol precursors, with the effect of changing the aerosol mix optical properties, hygroscopicity, number, and size. I found the manuscript and its results interesting, and generally well written and clear. I recommend that some aspects are clarified before publication.
> p. 3, 19 – Could you give a few more details on what are the actual dust-pollution interactions happening in the model? i.e. which are the other aerosol species interacting with dust? Can you specify what "pollution" is exactly here? Black carbon?

We have expanded section 2 to better present the coagulation of pollution and dust particles and the condensation of gaseous pollutants on mineral dust as the main dust-pollution interactions. "Pollution" is defined based on the emission sources as described in section 3, most of the black carbon is part of pollution but some fraction originates from natural biomass burning.

> p. 3, 24 - Please discuss dust optical properties in your model in light of the relevant literature (e.g. Kaufman et al. 2001, Müller et al., 2011, Di Biagio et al. 2019). It is elevant to assess this potential source of uncertainty.

We have added a discussion on the imaginary part of the refractive index of dust and included the references.

> p. 3, 34 - The indirect statement about the particle size distributions seems unjustified. AOD at a given wavelength is usually parameterized as the product of aerosol column loading for a given size mode/bin times the size-dependent mass extinction efficiency. At least, the statement should be supported by additional evidence that the optical properties that you use are "reliable".

We are aware that these observations are no proof but only an indication of a realistic size distribution. To give this better justification, we have rephrased the statement and now mention a reliable ratio of the extinction efficiency as a precondition for the argument. Still, we do not expect a big uncertainty in this ratio because of the consistent treatment of the optical properties throughout the spectrum.

> p. 5, 11 – Is there a spatial component in your procedure of reducing anthropogenic biomass burning emissions?

No, since we do not have more detailed estimates, a constant fraction is used globally.

> p. 5, 20 – Can you elaborate on the two different strategies for averaging?

From the SST results we use annual global mean values as "x" in Eq. (2) whereas for the nudged simulations we skip the global averaging and apply the equation for each grid cell separately to obtain the spatial distribution of the interaction term. We clarified the formulation.

> p. 5, 25 – Please clarify what your strategy is, in light of e.g. IPCC AR5 terminology: "The difference of the radiative fluxes from both calls yields the instantaneous forcing . . ." vs e.g. "Globally averaged, the net forcing in the solar spectrum shown in Fig. 2 (a) is (0.23 ± 0.01) Wm−2, the SST simulations yield an ERF of (0.3 ± 0.1) Wm−2" (p. 7, 8). Please clarify how each metric that you will discuss is calculated, and try to be consistent in the terminology you chose, in differentiating between instantaneous RF, ERF, and direct and indirect radiative effects.

We have expanded the paragraph to provide more a detailed presentation of the forcing definitions and calculations.

> p. 6, 6 – In light of your strategy (i.e. experimental design) is that because of a local aerosol effect or else?

Since the locations of regional maxima of this aerosol effect (other effects cancel in Eq. (2)) resemble the pollution hotspots over Asia, Europe and North America, the effect is clearly dominated by the local aerosol. Nevertheless, transport is involved as well, in particular in bringing desert dust to non-arid regions (e.g., Europe).

> p. 8, 17 – "Our SST simulations . . ." please rephrase, and resume briefly your experimental setup in the conclusions. Also, in this section you should discuss the limitations related to the experimental setup, as suggested by the editor.

We have rephrased the sentence and expanded the first paragraph of the conclusions by a brief summary of the experimental setup and a discussion of limitations.

> Figures 1, 2, 3 - It would be useful to see the individual (dust, pollution) effects alongside with their interactions. It would also be useful to see the average dust and pollution burden maps somewhere also within this manuscript.

We have added the new figures S2 to S9 to the supplement, showing the dust and particulate pollution burdens as well as the individual effects of dust and pollution on clouds and radiation.

> Figure captions – "Over stippled regions the results are consistent with zero at 2 $\sigma$ significance level." What do you mean by "zero"? A null hypothesis of zero difference in the mean among the ensemble members? Please clarify the captions, and provide the details of your procedure.

The null hypothesis is no effect, sigma is estimated by the standard error of the mean of the annual values (page 5, line 4ff). We have rephrased to "Over stippled regions the results differ from zero by less than two times the SEM of the annual values."

> Figure 4 – The bar chart is not very clear in its present form. Please make it explicit which simulation ensembles you are depicting in each of the top bars: only pollution and dust+pollution?

We now mention explicitly the terms represented by the bars.

> References
> Kaufman, Y. J., D. Tanré, O. Dubovik, A. Karnieli, and L. A. Remer, 2001: Absorption of sunlight by dust as inferred from satellite and ground-based remote sensing. Geophys. Res. Lett., 28, 1479–1482, https://doi.org/10.1029/2000GL012647.
> Müller, T., A. Schladitz, K. Kandler, and A. Wiedensohler, 2011: Spectral particle absorption coefficients, single scattering albedos and imaginary parts of refractive indices from ground based in situ measurements at Cape Verde Island during SAMUM-2. Tellus, 63B, 573–588, https://doi.org/10.1111/ j.1600-0889.2011.00572.x.
> Di Biagio, C., Formenti, P., Balkanski, Y., Caponi, L., Cazaunau, M., Pangui, E., et al. (2019). Complex refractive indices and single scattering albedo of global dust aerosols in the shortwave spectrum and relationship to iron content and size. Atmospheric Chemistry and Physics Discussions, 19(24), 15,503–15,531. https://doi.org/10.5194/acp‐2019‐145.

---

## Author Comment (AC2) · 30 Sep 2020

**Reply to RC2**

We thank the reviewer for the encouraging and very helpful comments. Below please find the point by point reply to the comments.

> This manuscript describes global model simulations to assess the impact of dust pollution interactions in cloud liquid and ice water content and direct and indirect aerosol radiative effect. The design of numerical simulations is appropriate and methodology to delineate various terms is reasonable. The results indicate dust aerosol interaction will reduce the negative cooling effect of aerosol. The result is quite interesting and worth publication. However, some clarifications of the methodology are required to facilitate better understanding.
> P3, L28-29, The sentence reads awkward.

We have rephrased the sentence.

> The methodology section, it would be better to move the description of four experiments with ('0', 'dust','Anthropogenic','Full') first. Then describe the 16 ensemble simulations and the nudged simulations for each experiment.

We have changed the order as suggested.

> Table 1-2 and 3S-4S caption, I am very confused here. The Mineral dust and Anthropogenic pollution by their definition, either only contains dust or pollution and there should be no dust pollution interaction, why do these simulations include the interactive term? Shouldn't 'Mineral dust' effect be simply Xdust – Xo and "Anthropogenic" effect be Xpol – Xo and the last term computed with Eq. 2?

$x_{\mathrm{pol}} - x_0$ yields the effect of the pollution, but in the absence of mineral dust (or ignoring the dust-pollution interactions). Due to the dust-pollution interactions this differs from the pollution effect in the presence of dust, $x - x_{\mathrm{dust}}$. We chose to present the latter because it represents the change from preindustrial times to the present day, wheras the dust free case is only a hypothecial scenario. However, the interaction term $\Delta_{\mathrm{int}}x$ provided in the last column immediately relates both effects, as $x - x_{\mathrm{dust}} = x_{\mathrm{pol}} - x_0 + \Delta_{\mathrm{int}}x$. Consistently, we also present the dust effect in the presence of pollution, $x - x_{\mathrm{pol}}$.

> P5 L23-24. What do you mean by neglecting 'aerosol-radiation-interaction'? Do you mean by excluding the aerosol contribution in the radiation calculation? That should not be termed as "aerosol-radiation-interaction".

We used the term "aerosol-radiation interactions" (ari) consistently with AR5 for the scattering and absorption by aerosol particles. Ignoring these interactions corresponds to excluding the aerosol contribution in the radiation calculation (but still including the result of the aerosol-cloud interactions (aci)). To avoid any confusion with other interactions subject of this article we now explicitly refer to "scattering and absorption by aerosols".

> P5 L28. How do you compute the total radiative forcing? Do you mean "total aerosol radiative forcing"?

Yes, we have revised to "total aerosol radiative forcing". More details on the forcing calculation have been included in the preceding paragraph.

> Section 5. How do you define the TOA radiative forcing in SW and LW? Is this simply the difference of reflective SW and outgoing long wave (LW)? What direction is considered positive, into the Earth or out of Earth?

We follow the convention to define incoming (downward) radiative fluxes to be positive and outgoing (upward) fluxes to be negative, which we now mention in section 3. The forcings correspond to the change of the net flux, the sum of incoming and outgoing fluxes. The SW forcing only includes in- and outgoing SW radiation, the outgoing LW radiation is included in the LW and the total forcing. We have clarified the second sentence of the section.

> P7 Last paragraph and Figure 4: it contains a lot of calculations that are not straitforward to readers. It took me a while to figure out (hopefully I got them correct!). Better to spell out how each term is calculated. For example, $X_p - X_o$ represents total aerosol effect without dust and $X - X_o$ represent total aerosol effect with dust (blue + green). The green part is computed from the difference of two calls of radiative transfer code with or without aerosol contributions. Then the blue part is total minus the green part. The red bar should be result of Eq. (2). Why is this number different from the global total in Figure 3?

We now provide the terms used for "With dust" and "Without dust" in the caption and have expanded the description of the forcing calculation in section 3. Figure 4 shows the effective radiative forcings (ERFs) obtained from the SST simulations, Fig. 3 shows results from the nudged simulations.

> I haven't understood the rational for using maps from nudged simulations while global averaged effect (Table 1 and 2) from SST simulations in the main article. Could you explain how each of these simulation configurations contrast and complement with the story you which to tell?

By definition, the effective radiative forcings are obtained from SST simulations which we now explicitly mention in section 3. Also in section 3, we discuss that due to statistical variability the SST results are not suited for regional analysis and we have to resort on nudged simulations for that purpose. Still, we provide global averages from the nudged simulations to show that SST and nudged results are largely consistent.